# Prioritizing Community-Based Intervention Programs for Improving Treatment Compliance of Patients with Chronic Diseases: Applying an Analytic Hierarchy Process

**DOI:** 10.3390/ijerph18020455

**Published:** 2021-01-08

**Authors:** Do Hwa Byun, Rho Soon Chang, Myung-Bae Park, Hyo-Rim Son, Chun-Bae Kim

**Affiliations:** 1Korean Association for AIDS Prevention, Gangwon Branch, Chuncheon 24405, Korea; byeondohwa@daum.net; 2Department of Public Administration, Kangwon National University, Chuncheon 24341, Korea; rschang@kangwon.ac.kr; 3Department of Gerontology Health and Welfare, Pai Chai University, Daejeon 35345, Korea; parkmb@pcu.ac.kr; 4Hongcheon County Hypertension and Diabetes Registration and Education Center, Hongcheon 25135, Korea; hc_health@naver.com; 5Department of Preventive Medicine, Yonsei University Wonju College of Medicine, Wonju 26426, Korea

**Keywords:** hypertension, diabetes, treatment compliance, community-based intervention program, analytic hierarchy process

## Abstract

The purpose of this study was to apply multicriteria decision making and an analytic hierarchy process (AHP) model for assessing sustainable management of hypertension and diabetes. Perception of two alternative health care priorities was also investigated. One priority was improving treatment compliance of patients with hypertension and diabetes. The other priority was strengthening the healthcare system for continuous care. Our study design to evaluate community-based intervention programs for hypertension and diabetes was developed using brainstorming, Delphi techniques, and content analysis along with literature review. We finally proposed a hierarchical structure of the AHP model with 50 third sub-criteria in six levels. By surveying this AHP questionnaire to a total of 185 community health practitioners in Korea, we found that improving treatment compliance of patients with chronic diseases should be relatively more important than strengthening the healthcare system. Further research is needed to expand survey subjects to primary care physicians and even policymakers of central government for the appropriate application of this AHP model.

## 1. Introduction

The Global Burden of Disease (GBD) Study 2013 reported a substantial (42.3%) increase in years lived with disability (YLD) from 1990 to 2013. This was because, in 2013, chronic diseases were overwhelmingly dominant in the top 20 leading causes of YLDs worldwide, without any infectious diseases [1]. Between 1990 and 2013, numbers of deaths from chronic non-communicative diseases and injuries steadily increased while deaths from communicable, maternal, neonatal, and nutritional causes decreased. For most of the leading non-communicable diseases (NCDs), the number of deaths had increased by 42% from 27.0 million (UI 26.3–27.6) in 1990 to 38.3 million (37.2–39.4) in 2013 [2]. Among chronic diseases, hypertension and diabetes mellitus were major causes of mortality not only in most of developed countries, but also in developing countries with high-burden, requiring disease management and continuous treatment [3]. For these reasons, monitoring GBD for realizing UN Sustainable Development Goals (SDGs) has become an important consensus at the World Health Organization (WHO) [4]. However, within the public health system of most countries, many hypertension and diabetes mellitus patients do not comply with treatments such as regular and continuous revisits to health care facilities and taking medicine. Their conditions become worse due to their old age and comorbid infection such as the coronavirus disease-2019 (COVID-19). They often suffer complications and die. A meta-analysis by Huang et al. [5] has recently reported that diabetes is associated with mortality, severity, and disease progression in patients with COVID-19.

### 1.1. Chronic Diseases Intervention Program and Rule of Halves

One cornerstone of community-based intervention programs for early detection and treatment of chronic diseases including hypertension began in the early 1970s, continuing up to date in Finland [6,7,8,9]. To be related with this project, classic studies the “rule of halves” for controlling hypertension in the community had been published (i.e., only half of hypertensive population were undetected, half of those detected were untreated, and half of those who were treated were not controlled for their hypertension) [10,11]. Although various efforts have been made for the detection and prevention of hypertension globally, recent population studies have shown that awareness and management of high blood pressure (BP) levels are far from optimal [12]. Empirically, Marques-Vidal et al. [13] have performed a systematic review and indicated that, although the “rule of halves” is no longer applied for screening and treating hypertensive patients in industrialized countries, it might still be valid for early detecting and caring for hypertension in developing countries and for effective antihypertensive drug treatment in all countries.

In South Korea, due to its rapid economic growth and aging since 1990s, chronic diseases such as hypertension and diabetes instead of infectious diseases have surged to bring a huge national medical cost. Amid epidemiological transmission, this rule of halves has also been shown to be valid in Korea’s community-based intervention program for controlling hypertension [14]. According to the Korea Health Statistics 2018, among Korean adults aged 30 years and older, hypertension awareness rate increased from 66.3% (during 2007–2009) to 69.1% (during 2016–2018), rising by 2.8%. Hypertension treatment rate also improved from 60.3% to 65.3%, rising by 5.0%. The rate of controlled hypertension among treated population was 1.5 times higher than that among the entire hypertensive population (73.1% vs. 48.3%), showing an upward pattern [15].

Today, addressing chronic diseases is a major challenge for healthcare systems in all countries. These healthcare systems have largely been developed to cope with acute episodic medical services rather than to provide well-organized care for people with chronic conditions [16]. Control of NCDs in primary care plays a key role in both prevention and treatment for elderly patient with chronic conditions. There is a need to understand which interventions are effective, for whom, and in what context. Defining features and delivery stage of primary health care (including registration, continuity, coordination, and comprehensiveness) is important for controlling chronic conditions [17]. In the 1990s, Wagner et al. [18] developed a chronic care model (CCM) as a platform to enhance the quality of chronic care. It was an organizational approach to caring for people with NCDs. CCM is particularly applicable in a primary care setting. Reynolds et al. [19] have confirmed that self-management support is the most frequent intervention among six elements of the CCM. It is significantly associated with continuous management, predominately for diabetes and hypertension.

Recognizing this, after establishing a comprehensive plan for cardio-cerebrovascular diseases in 2006, the Ministry of Health and Welfare chose Daegu Metropolitan City as the pilot region to register and manage hypertension and diabetes patients through local clinics and pharmacies in 2007 in order to establish the management for hypertension and diabetes not only as an individual approach, but also as a community-centered management system. This chronic disease management intervention (Korea Hypertension Diabetes Daegu Initiative, KHyDDI) first began four years ago in a rural area. It is foremost now within the main framework of CCM [20,21]. Afterwards, registration and education centers for hypertension and diabetes in 2012 were expanded to 19 districts of 12 Metropolitan cities and Provinces [22] with budgets allocated from central and local government as shown in Figure 1. Unlike the central government’s original plan, however, these community-based intervention programs are currently only operated in 9.8% of the country, with additional participation from some local governments [23,24].

Since 1989, the Korean government has already introduced ambitious universal coverage schemes via a national health insurance system based on fee-for-service, moving significantly faster than industrialized countries [25,26,27]. Unlike the government-led chronic disease management project, the National Health Insurance Service has conducted a “medication counselling intervention program” or a “community based primary care project” since 2012 as a chronic disease management program, focusing on insurer-led or (primary care) physician-led management of hypertension, hyperlipidemia, or diabetes [28,29]. Despite this universal health coverage, a nationwide survey has found that there are still socioeconomic disparities in the control of patients with hypertension and diabetes in Korea [30,31].

### 1.2. Chronic Diseases Management Policy and Analytic Hierarchy Process (AHP)

Treatment or medication adherence is well known to be related to continuity of care, which might render healthcare providers more accountable for managing their patients with chronic diseases continuously [32]. Continuity of care is defined as continuous and consistent supply of medical services by healthcare manpower to meet patients’ medical needs or the concept of a “continuous caring relationship” between doctor and patient [33,34]. Continuity of care is especially important for the effectiveness of chronic disease management [35]. Primary care services for controlling chronic diseases need to make active efforts to minimize diverse obstacles to continuity of care in a community. Considering payment methods for a physician, out-of-pocket fee-for-service payment is a common deterrent not only to access but also to continuity of care [36,37]. With regard to comprehensiveness, continuity, and patient-centered care, private clinics do not perform differently than public health centers. In rural areas rather than urban areas of most countries, governmental health centers are usually designed to work in close relationship with the community they serve [38]. Considering a rapidly aging population and the increased prevalence of chronic diseases in Korea, the new administration led by Moon Care since 2017 is willing to reform health care system toward moving a primary care orientation [39]. However, without a comprehensive review of the primary health care policy for chronic disease management, there is room for confusion. Thus, it is important to secure clear evidence necessary for policy decisions and priorities.

The AHP method was designed by Saaty in the late 1970s to aid managers in decision-making of issues about qualitative, quantitative, and mixed criteria. The AHP was proposed to deal with both instinctive and rational decisions when selecting the best priority from two alternative policies according to multicriteria [40,41]. Using the AHP method in decision-making involves a three-phase process, including hierarchical building phase, performing paired comparisons and calculating weights, and finally determining system compatibility [42].

Several research studies have proven that the application of AHP to support group decisions is useful in medical and health care fields [43,44,45,46,47,48,49,50]. Since 1981, AHP has been applied inconsistently in healthcare research. However, using AHP for evaluating health care has increased since 2005 [51]. According to results of a systematic review, AHP is a promising support method for shared decision-making between patients and physicians, evaluation and selection of diagnoses and treatments, and evaluation of health care technologies and policies [52].

Therefore, the aim of this study was to implement a comparative analysis for sustainable management of hypertension and diabetes and investigate the perception of the relative importance of two alternative healthcare priorities using an AHP model. One of these two alternative approaches was that a community-based intervention program would improve treatment compliance of patients with chronic diseases at the individual level. Another alternative was that the program would enhance the continuity of care at the community level.

## 2. Materials and Methods

### 2.1. Developing the Analytic Hierarchy Process (AHP) Model

We applied the AHP model as an evaluation model based on priority selection for continuous management of hypertension and diabetes in community-based intervention programs. According to Saaty’s axioms for AHP [40,42], we structured the AHP model largely in three steps as suggested by Shin et al. [46] and Dolan et al. [47] as follows: (1) preliminary phase—systematic review on chronic disease management programs, content analysis of annual reports of the Hypertension and Diabetes Registration and Education Center (*HyDiREC*), and brainstorming among Hongcheon Center’s staff for establishing an evaluation model; (2) new formulated phase for decision hierarchy—define decision elements, construct decision model, and decompose decision into smaller parts; (3) AHP questionnaire survey and priority analysis phase—perform a nationwide survey with AHP questionnaire and compare importance of criteria and the alternatives’ priorities.

At the first AHP step, theoretical and prior researches related to main interest of this study, i.e., chronic disease management programs, were reviewed [6,7,8,17,33]. First of all, theoretical models related to treatment compliance of patients with chronic diseases included a health belief model [53], Andersen’s medical care utilization model [54,55], and the health lifestyle paradigm [56]. Results of 32 articles (including Korean articles) [6,7,8,10,11,12,13,14,17,18,20,21,57,58] were collected and divided into domains (or variables) of each theoretical model to identify effects on treatment compliance and draft an evaluation model. Up to 2014, 428 articles on continuity of care and health care system were collected through literature search using databases such as PubMed. Among them, 50 articles, including those from Australia [59], Italy and Sweden [60], Canada [61], UK [62], Taiwan [63], Finland [64], and Korea [65], were reflected in the elaboration of the initial evaluation model. The AHP evaluation alternatives about treatment compliance at patient level and continuity of care at community level was currently associated with the “chronic care model (CCM)” and “health system strengthening (HSS)” emphasized by main international organizations [66,67,68]. This also coincided with the concept of health promotion or lifestyle modification for prevention of chronic disease from WHO [69]. In addition, our research team reflected results of content analysis conducted on the 1st annual report of hypertension and diabetes registration management program in five regions (Hongcheon-gun, Gangwon-do; Jung-gu, Ulsan; Bucheon-city, Gyeonggi-do; Jeju-city, Jeju-do; and Gwangsan-gu, Gwangju). In this stage, the basic framework of the AHP analysis model, which consisted of infrastructure (budget, workforce input)—process (implementation of program)—outcome (participation of health care facilities, patient registration, counselling, and completion status of education) suggested by Shin et al. (2009) [46], was utilized.

### 2.2. Decision Hierarchy Criteria for Selecting Prioritization

In the second AHP step, the aim of this study was defined following the main criteria that could be divided further at lower levels into sub-criteria. This stage began with brainstorming of a group of experts consisting of seven healthcare practitioners in Hongcheon-gun *HyDiREC* and a regional public health center and four academic professionals. Qualified healthcare practitioners such as nurses, nutritionists, or exercise instructors were mainly responsible for patient registration, health education (about diseases, nutrition, and exercise), monitoring and counseling to patients, and links to health care facilities within the center. Academic professionals majoring in preventive medicine, health science, and administration also participated in this study while supporting the planning, implementation, and evaluation of the Center’s annual work. In principle, brainstorming requires members to initially state as many ideas as possible, including improving or merging previously stated ideas [70].

Summing up results of the previous phase, our decision hierarchy finally consisted of six levels: a goal, four levels of criteria, and alternatives. These elements are represented in a tree-like structure. The hierarchy represents the structure of the decision problem and forms the basis of comparisons that have to be made in the following phases. In this study, the primary goal was to choose an alternative for sustainable management of hypertension and diabetes mellitus in the community. The hierarchy includes four levels of criteria (main criteria, first sub-criteria, second sub-criteria, and third sub-criteria). In our study, the two major evaluation criteria for sustainable management of chronic diseases were defined as approach strategy for patients (individual level) and healthcare system (community level), respectively. The approach strategy for patients consisted of health behavior and behavior of health care utilization in the first sub-criteria. The first evaluation sub-criteria for approach strategy of healthcare system were classified as program infrastructure, program process, and program outcome.

The program infrastructure stage evaluated personal, physical, and financial resources needed to implement a program. A strong infrastructure for community-based intervention program is critical to assure the registration of patients with hypertension and diabetes mellitus, to conduct their education programs, to build up the public-private partnership, and to link with community resources (primary care physicians, pharmacists, senior community health volunteers). It also requires adequate government funding. In this process stage, we evaluated the feasibility and adequacy related to program management and implementation that could be used to scale up a program quality both immediately and in the future. Finally, the outcome stage of program evaluation determined the efficiency of spreading the program into other regions. This includes an assessment of the registration rate in the regional *HyDiREC*, regular visiting rate in their community clinics, the adequacy of the health education reports, and the timeliness and up-to-date status of the program.

In this AHP model, core factors of the second sub-criteria consisted of six major categories: (1) input factors for building infrastructure, (2) participation mechanism within a community, (3) adequacy of center operation and provision of services, (4) adequacy of administrative steps, (5) program output, and (6) program performance. The third sub-criteria in the fifth level of the hierarchy were then divided into 50 critical factors.

Given the hierarchy illustrated in Figure 2, the priority setting procedure began by pairwise comparison of the criteria with respect to the overall goal.

### 2.3. AHP Questionnaire Survey and Priority Analysis

In the last AHP step, we investigated the validity of our criteria and the hierarchical structure of this AHP model from 1 April 2014 to 7 April 2014. The Delphi method was also used in this phase to design AHP questionnaires completed by a few expert groups in other *HyDiRECs* in Korea [22,23]. As a result, the AHP survey questionnaire was ultimately confirmed through revision by performing the pilot survey for seven staff of Hongcheon-gun *HyDiREC* and four professionals associated with this community-based intervention program. The AHP questionnaire was composed of basic information for respondents and relative importance evaluation based on the Likert scale (7-point scale, 2 main parts) and priority evaluation through pairwise comparison by criteria (9 points in both directions). To aggregate survey subjects’ weightings, respondents were asked to rate each group weight using the 7-point Likert scale from “not very important” to “very important.” The AHP survey questionnaire had alternative options of pairwise comparison composed of “improvement for treatment compliance” and “enhancement for continuity of care.” The numerical scale used to make pairwise comparison ranged from 9 to 1 (9 points in both directions), with 1 denoting equal importance and 9 denoting the highest degree of importance. All criteria were compared in a pairwise manner when the AHP technique was used. AHP could also be used to measure scale properties (i.e., global weights) for elements in this model, which was referred to as prioritization [40,42]. Priorities (global weights) were measured with within-group analysis for all evaluation criteria in each level. Importance ranking was selected in the order of sequence of priorities value.

The AHP survey was conducted for a total of 185 community health practitioners working at 19 *HyDiRECs* and 19 regional public health centers nationwide in Korea. Survey questionnaires were collected through mail and e-mail. After fully explaining the content and the purpose of the research through mail or phone and obtaining the consent, the survey was conducted from 10 April 2014 to 3 May 2014. Among target subjects, 140 participated in this survey, showing a response rate of 75.7%. Finally, 133 sets of the survey questionnaire were used as database for analysis, excluding poorly answered ones. If the consistency ratio (CR) exceeded 0.2, respondents were excluded from the final analysis. If the CR value was less than 0.1 then it is considered “good,” if it was less than 0.2 then it is “fair” [40].

Data was further analyzed with a 90-day limited license of Expert Choice Desktop for academy (EC) for AHP evaluation [71]. Right now, the above version is no longer supported and another demo version with same function is provided on this website. In addition, descriptive analysis such as average and frequency was performed for each main characteristic of AHP survey participants. The geometric mean was calculated by averaging participants’ responses at each point of importance comparison to form a composite matrix using SAS 9.2 for Windows (Institute Inc., Cary, NC, USA). For the arithmetic mean (*Mg*), the sum of each mean in the evaluation criteria (options) between levels, as shown in the following formula, was 1. However, the geometric mean was calculated with the formula of ‘log*Mg*’. Since weights analyzed by AHP were values with a decimal point of 1 or less, the sum of those means was less than 1. Nevertheless, the use of geometric means was not an issue when it came to ranking comparisons measured for relative importance [72].
$$Mg=\sqrt{X_{1}\times X_{2}\times X_{3}\times \cdot \cdot \cdot \times X_{n}}\;$$
$$logMg=\overset{N}{\underset{i=1}{\sum }}\log Xi/N$$

## 3. Results

### 3.1. General Characteristics of AHP Survey Participants

Table 1 shows results of descriptive analysis for general characteristics of 133 subjects and 70 valid respondents. Among participants, 17 (12.8%) and 116 (87.2%) were men and women, respectively. The age distribution of those surveyed was follows: 42.1% (56) in their 30s and 29.3% (39) in their 40s. In terms of academic background, 97 participants (72.9%) graduated from universities, and 14 (10.5%) and 13 (9.8%) had master’s and doctorate degrees, respectively. Regarding their majors, nursing and nutrition accounted for 72 (54.1%) and 30 (22.6%), respectively. This is because nurses and nutritionists are required in the workforce guideline of the *HyDiREC* in Korea Centers for Disease Control and Prevention (KCDC). Considering that most directors in the *HyDiREC* were university professors, 78 (58.6%) and 40 (30.1%) participated in the *HyDiREC* and each regional public health center, respectively. Their average work experience in the field was 69.8 months or 5 years and 10 months. The valid response rate excluding 63 subjects from the final analysis was 52.6%, depending on the value of the CR. The frequency by general characteristics of a total of 70 valid respondents was similar without showing significant difference from the frequency of survey participants (133).

### 3.2. Importance Evaluation to Approach Strategy for Patients and Healthcare System

In this section, we present the results of relative importance evaluation by a 7-point Likert scale. As a result of the AHP model analysis, when the importance evaluation among criteria for sustainable management of hypertension and diabetes was compared in terms of patients and healthcare system dimension, the approach strategy for patients was found to be more importantly perceived. Healthy life-style (anti-smoking, reducing alcohol drinking, nutrition, physical activities) (6.26 points), risk awareness for hypertension and diabetes (likelihood and severity of disease) (6.23 points), and health examination participation rate (blood pressure, blood glucose) (6.14 points) were included at higher ranks of the importance at patients (individual) level. These all belonged to the domain of health-related behaviors (Table 2). Health practitioners in 19 regional public health centers of Korea seemed to view patients’ health-related behaviors (at individual level) more importantly given clinical features of hypertension and diabetes.

On the other hand, the specialty of manpower (6.27 points), working continuity of manpower (6.23 points), adequacy of health education content (6.19 points), and adequate selection and disposition of manpower were included at higher ranks of importance at healthcare system (community) level. All these sub-criteria belonged to the domain of appropriateness of center operation and service provision under the program process evaluation (Table 3). This finding indicates that the appropriate health education by specialized staff and the service provision guaranteed by working continuity of staff are recognized as important factors for sustainable management of non-communicable diseases in local communities. Thus, registration and education centers for hypertension and diabetes can further strengthen the current health care system in Korea.

### 3.3. Pairwise Comparison of Evaluation Criteria for Community-Based Intervention Program

In our study, results of the AHP modeling were divided into criteria evaluation and relative priorities of decision alternatives for sustainable management of hypertension and diabetes. Criteria evaluations for sustainable management of hypertension and diabetes were first divided into four stages according to hierarchical criteria. The priority awareness toward evaluation criteria for sustainable management of hypertension and diabetes is shown in the following.

First, as shown in Table 4, the evaluation of approach strategy for patients (0.525) was more important than the approach strategy for healthcare system (0.332) in the main evaluation criteria. This indicates that even if the healthcare system strengthening strategy at the community level is attempted through the hypertension and diabetes registration management program, which is being implemented based on chronic care model (CCM), the patient’s individual-level approach strategy for improving health behavior after recognition of the risk caused by the possibility and severity of chronic diseases including hypertension and diabetes should be prioritized.

Second, important factors of the first evaluation sub-criteria were the behavior of health care utilization (0.461) in the approach strategy for patients and the program infrastructures (0.295) in the approach strategy for healthcare system, respectively. In the second evaluation sub-criteria, important options were: (1) participation system within community (0.436) as one of program infrastructures, (2) adequacy of center operation and service provided (0.672) as one of program processes, and (3) program output (0.453) as one of program outcomes as shown in Table 4. This result shows that the “adequacy of center operation and service provided” is the most important factor of the intermediate evaluation criteria. This means services provided by the current hypertension and diabetes registration education center are appropriate and important for sustainable management of chronic diseases.

Third, the five most important factors of the third evaluation sub-criteria were: (1) patient satisfaction (0.0471), (2) healthy life-style (anti-smoking, reducing alcohol, nutrition, physical activities) (0.0393), (3) well-timed health policy (0.0353), (4) health examination participation rate (blood pressure and blood glucose test) (0.0343), and (5) risk awareness for hypertension and diabetes (likelihood/severity) (0.0318) (Table 4 and Figure 3). All these factors fell into the core sub-criteria of “the approach strategy for patients.” In terms of each of intermediate level evaluation criterion, the most important third evaluation factors were: (1) healthy life-style (0.0393) as part of health behavior, (2) patient satisfaction (0.0471) as part of behavior of health care utilization, (3) adequacy of governmental budget for program (0.0064) as part of investment resources for building infrastructure, (4) voluntary participation level of community residents (0.0115) as part of participation system within the community, (5) specialty (empowerment) of manpower (0.0073) as part of adequacy of center operation and service provided, (6) adequacy of repayment process of costs and observance of the demonstration program guidance (0.0034) as part of adequacy of administrative procedure, (7) pass rate of health education (about disease, nutrition, exercise) (0.0086) as part of program output, and (8) reducing the death rate of cardiovascular diseases (long-term effect) (0.0067) as part of program performance.

In this study, the patient satisfaction factor was found to be the most important criterion of all factors for sustaining the continuous management of hypertension and diabetes. Healthy life-style (anti-smoking, reducing alcohol, nutrition, physical activities) was the second most important factor. In the end, it was prioritized that the higher the patient satisfaction in health care utilization behavior and the healthier the lifestyle in health behavior, the easier it would be to improve treatment compliance for patients with hypertension and diabetes. This is of particular interest to healthcare managers and policymakers who currently have limited access to evidence-based health policies and would like to learn how to implement and understand the community-based intervention program within their scarce resources.

### 3.4. Comparison of Alternative Preference and Priorities Gap According to Evaluation Criteria

These results illustrate that high-ranking indicators from first to third level evaluation criteria should be the first priority to successfully implement and manage a community-based intervention program for sustainable management of hypertension and diabetes. Based on final AHP results of our study, we can conclude that the improvement for treatment compliance is a better alternative than the enhancement for continuity of care (Table 5).

Next, this study performed a preference gap analysis between improvement for treatment compliance and enhancement for continuity of care according to the evaluation criteria. The purpose of this analysis was to specify how much more preferable the improvement for treatment compliance was compared to the enhancement for continuity of care with respect to each criterion. That is, judgments regarding the relative preference of the alternatives are made relative to each criterion. Table 6 shows priorities of alternatives and absolute values of differences (gap) between priorities of alternatives in terms of the evaluation criteria.

First, the “improvement for treatment compliance” alternative was only preferred to the “enhancement for continuity of care” alternative at the approach strategy for patients in the main evaluation criteria. However, the “enhancement for continuity of care” alternative was preferred to the “improvement for treatment compliance” alternative only at the approach strategy for healthcare system. Second, when evaluating alternatives using the first sub-criteria, the “improvement for treatment compliance” was preferable to the “enhancement for continuity of care” particularly for health behavior, health care utilization behavior, and program outcome. Finally, when evaluating alternatives using the second sub-criteria, the “improvement for treatment compliance” was preferable to the “enhancement for continuity of care,” particularly for investment resources for building infrastructure, participation system within community, and program output and program performance. This means that the “improvement for treatment compliance” makes a larger contribution than the “enhancement for continuity of care” with respect to achieving program goals. The most contributive factor in our study was defined as the factor with the largest gap. The five most contributive factors of intermediate level evaluation criteria were: (1) health behavior, (2) health care utilization behavior, (3) program outcome, (4) investment resources for building infrastructure, and (5) program output.

## 4. Discussion

First of all, this study is the first attempt to assess priorities at the level of patient’s individual and healthcare system of community-based intervention programs for sustainable management of chronic diseases. Management of chronic diseases is not just a problem in developed countries today [2,3]. Chronic diseases are directly related to aging due to their characteristics, but they are also largely related to how each patient usually maintains health-related behaviors before the age of 60 [12,13,14]. To date, diverse interventions have been applied along with various theoretical models to solve this problem at national or community level in both developed and developing countries. Korea is currently trying to quickly respond to policy suggestions through implementation and expansion of community-based intervention programs in the primary care setting in a national level [19,21,22,23,24,28,29]. It is important not only to find hidden hypertension and diabetes patients within the community, but also to find patients who do not continue treatment early [10]. The optimal implementation strategy of community-based intervention program for sustainable management of hypertension and diabetes depends on the presence of such an evaluation mechanism. This study developed an evaluation model to find the best alternative for sustaining the continuous management of hypertension and diabetes using the AHP as the most promising method for prioritizing requirements as China’s case [58]. In this study, results of AHP modeling were divided into the evaluation of criteria and relative priorities of decision alternatives of community-based intervention for sustainable management of chronic diseases.

Continuous pursuit of management for controlling chronic diseases, whether it is led by the government or the insurer [24,28], is also important. Still many hypertension and diabetes patients are either not managing their own diseases well or are not aware of this disease. Especially, among total geriatric patients, 24% have reported the experience of ceasing continuous treatment. Among those, 50% had financial reasons [57,73]. Here, we have to understand that “treatment adherence” and “continuity of care” for patients with chronic diseases can be both sides of a coin. Conceptual models of barriers to adherence to the elderly describe patient, prescriber, and health care system factors. Some potential barriers (i.e., factors associated with nonadherence) in older adults have been identified in a few studies [73,74,75], including patient-related factors (such as disease-related knowledge, health literacy, and cognitive function), drug-related factors (such as adverse effects and polypharmacy), and other factors including the patient-provider (or prescriber) relationship and various logistical barriers to obtaining medications. Furthermore, non-adherence with prescribed antihypertensive medications is only one aspect of a patient’s attitude towards healthcare delivery. Patients may display additional features that could be deleterious, such as non-adherence with healthcare appointments. They may pursue self-destructive lifestyles such as unhealthy diets and physical inactivity [76]. Roy et al. [77] have identified the following facilitating and inhibiting factors that can affect continuity of care to physicians: (1) physician-patient relationship, interest in clinical continuity of care activities, positive role models, working alongside a nurse, and adequate access to community resources as main facilitating factors, and (2) scope of administrative duties, interest in a comprehensive practice, a negative experience of continuity of care during training, a sense of inadequacy with respect to continuity of care, a heavy case load, and a lack of support in the practice as main barriers. This ambivalence was also found in the analysis results of the priority perception according to AHP evaluation criteria developed in this study.

Our researchers, who had participated in the planning and evaluating Hongcheon-gun Hypertension and Diabetes Registration and Education Program for the past eight years [78,79], finally selected two opposing views of community-based intervention programs as policy alternatives to the AHP evaluation model that faced new challenges amid drastic changes in health policies in Korea. Unlike other AHP studies [43,44,45,46,47,48,49,50,51,52], this study largely had a three-step mixed process. We developed AHP evaluation criteria using research methods such as extensive literature review, brainstorming, content analysis, Delphi technique, and AHP questionnaire survey (including the pilot survey) based on the long experience of carrying out community health projects in Ganghwa-gun, Korea, and Tikapur, Nepal [80]. Therefore, our research team expected that the dimension for strengthening the health care system would take precedence over a patient’s health-related behaviors. However, our results showed the opposite. That is, current health practitioners recognized that the approach strategy for patients (0.525) for improving treatment compliance was very important in the main evaluation criteria. This was partly because the surveyor was limited to community health practitioners who directly participated in the current hypertension and diabetes registration and education project in Korea. Of course, the analysis of priority perception showed that “healthcare utilization behavior” (0.461) in the patient’s individual domain had the higher priority than “program infrastructure” (0.295) in the healthcare system. In second sub-criteria, “adequacy of center operation and service provision” (0.672), “program output” (0.453), and “community participation system” (0.436) each ranked the highest in priority. Moreover, the “patient satisfaction” factor (0.0471) and “healthy life-style (anti-smoking, reducing alcohol, nutrition, physical activities)” (0.0393) were found to be the most important criterion of all evaluation criteria factors in third sub-criteria. This result was the same as those of studies on preference of patients using AHP for medicinal therapy for type 2 diabetes [81,82]. Therefore, it is necessary to emphasize the specialty of health behaviors and lifestyle modification from the planning stage of future community-based intervention programs. It is also important to provide feedback to not only health practitioners, but also primary care physicians in the community in consideration of continuity of care [21,49,62]. Furthermore, the government should not only strengthen medical education on patients’ behavioral changes [77] such as with family physicians, but also actively promote the development of health insurance benefits package, such as in Thailand’s case [83].

Like our research, multicriteria decision analysis (MCDA) [40,72] is now being applied in various community health areas as an approach to promote evidence-based, patient-centered healthcare [47], prioritize health education needs for community participation [49], develop health intervention in the universal coverage health benefit package [83], and prioritize investment in public health intervention [84]. The need for supporting a patient’s behavior of healthcare utilization via system building was then seen as a priority for implementing the community-based intervention program. This suggests that for healthcare utilization behavior and management of non-communicable disease in accordance to health perception of an NCD patient, more emphasis should be placed on the importance of staff specialty that can provide proper services. On the other hand, in analysis results for priority perception according to third sub-criteria, patient satisfaction and healthy lifestyle ranked the highest priority, consistent with results of other reports [7,9,23,24,58] stating that in order to systematically treat and manage NCD patients, the central or local government should pay more active attention to prevention through securing budget and expanding public health care service.

This study makes several policy implications in relation to treatment compliance and continuity of care for sustainable management of hypertension and diabetes [85]. First, health education programs for hypertension and diabetes should be continuously provided to improve patients’ awareness on the significance of health-related behavior. This was based on results emphasizing patient level awareness for the continuity of hypertension and diabetes treatment done by professionals as target [86], including “awareness of risk concerning hypertension and diabetes (onset possibility/severity),” “healthy lifestyle (non-smoking, moderate drinking behavior, nutrition, exercise),” and “physical examination participation rate (checking blood glucose and blood pressure).” Second, the doctor-patient relationship should be reinforced by making the best use of the demonstration program where residents are encouraged to visit their nearest primary medical clinics for the continuity of care [87]. Unlike infectious diseases, in order to improve treatment compliance of chronic diseases such as hypertension and diabetes, a support from primary physicians to regularly visit the primary health care near the house is fundamentally important. Third, the government (central and local) should pursue a more evidence-based health policy [88]. The government will need to give positive support for strengthening the national health system of health policies for disease management so that patients can take care of themselves personally. Fourth, a systematic capacity building should follow constantly to secure the specialty of chronic disease management personnel. Fifth, the public-private partnership should come first based on the linkage with local community resources (for supporting economic incentives) as emphasized by the primary health care system [89,90]. As seen in result of this research, “the voluntary participation level of community resident” within program infrastructure as (public health system) domain was found to be important. Moreover, considering the “participation of private clinic and pharmacy,” which is the basic model for hypertension and diabetes registration and management assumed as precondition, an association with community resources (retired senior community health volunteers in case of Hongcheon-gun) [78] through establishment and utilization of public-private partnership within a community, is required for improving the efficiency in relation to program performance. At any rate, community-based intervention health programs with public-private partnership should be actively considered rather than unilateral projects led by the government or insurers to improve patients’ treatment compliance in the case of chronic disease management with CCM.

Although this study extracted and analyzed factors of the evaluation criteria on the relative importance for sustainable management of hypertension and diabetes, it had the following limitations. First, the response rate was high (75.7%), although contents of the questionnaire survey with the AHP method might be somewhat unfamiliar to community health practitioners at 19 sites, unlike other general surveys. However, the representation of health project participants nationwide in 254 public health centers and the intrinsic value of selective bias could not be ignored. Thus, there might be limitations in interpretation and utilization of our results. Second, our AHP questionnaire survey was not available to hypertensive patients, diabetes patients or the general public. This was because AHP questionnaire contents were very high in difficulty index and large in its quantity. Thus, answers to the questionnaire could not be easily written by these patients or the general public. Third, as a result of simplifying the hierarchy of evaluation criteria by focusing on consistency in the evaluation through prior systematic review and Delphi technique, there was a limitation in deriving concrete and in-depth analysis. Nevertheless, 70 out of 133 survey participants had CRs of 0.2 or higher, with a valid response rate of 52.6%. Another weakness of this study was that data from the 2014 survey were used for analysis, which might be somewhat less timely. However, the importance of chronic disease management can be emphasized as the political environment for health policy is rapidly changing due to the coronavirus disease-19 pandemic and recent increase of the AHP application in health and medical research.

## 5. Conclusions

We analyzed the relative importance of evaluation criteria using our proposed AHP model for intervention and management of chronic diseases in Korea. By evaluating the relative importance of the AHP model for sustainable management of hypertension and diabetes, all participants in our survey seemed to view patients’ preventive strategy as being important given clinical features of hypertension and diabetes. The domain of appropriateness of center operation and service provision in program process evaluation was also highly ranked as a healthcare system strategy. In conclusion, for sustainable management of hypertension and diabetes within regional communities, not only service provision through selection and continuous work of capable staff, but also the appropriateness of health education about health-related behavior that can improve a patient’s satisfaction and treatment adherence are important. Health education is only the first stage in lifestyle modification. Continuous health education programs for patients with diabetes and hypertension within a community are needed for their behavior change accordingly. Future research should focus on expanding survey subjects to primary care physicians and even policymakers of central government for a proper use of this AHP model.

## Figures and Tables

**Figure 1 ijerph-18-00455-f001:**
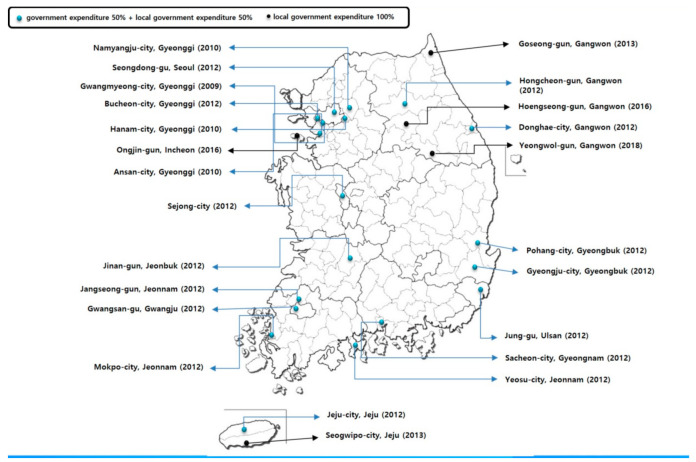
Current operation status of registration and education center for hypertension and diabetes in Korea. Years in parenthesis indicate the beginning time of the project.

**Figure 2 ijerph-18-00455-f002:**
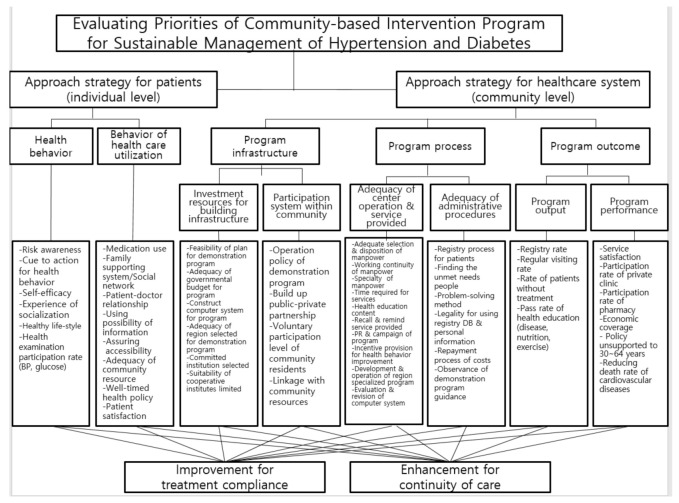
Hierarchical structure of community-based intervention program for sustainable management of hypertension and diabetes according to the AHP model.

**Figure 3 ijerph-18-00455-f003:**
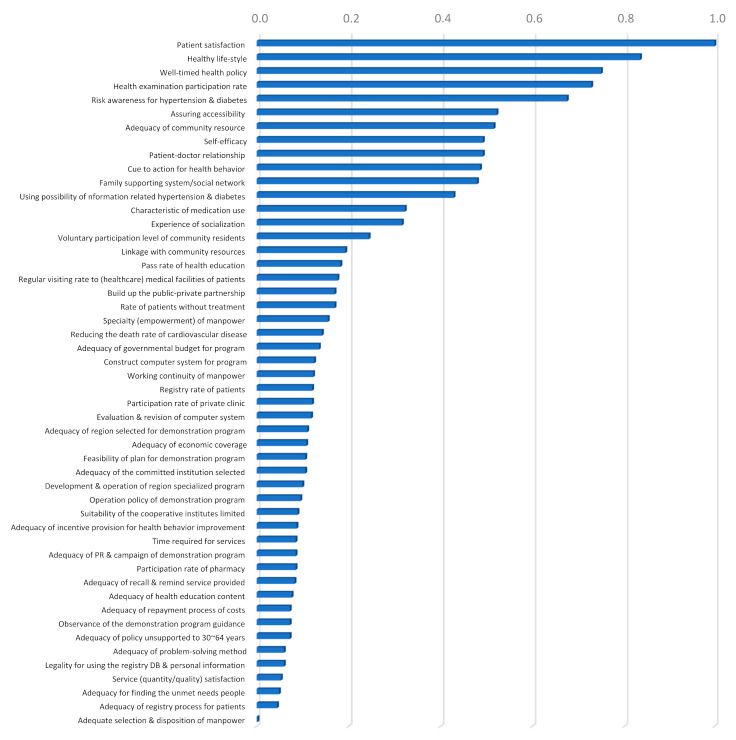
Priority of the third evaluation sub-criteria for community-based intervention program with pairwise comparison. To easily interpret priorities (global weights), the critical factor (patient satisfaction) with the highest priority was standardized to one. Other factors were rearranged according to their importance ranking.

**Table 1 ijerph-18-00455-t001:** General characteristics of analytic hierarchy process (AHP) survey participants. Unit: Person (%).

Characteristics	Category	AHP Survey Participants		Valid Respondents *	
		No.	%	No.	%
Sex	Male	17	12.8	12	17.1
	Female	116	87.2	58	82.9
Age group	20–29	20	15.0	14	20.0
	30–39	56	42.1	32	45.7
	40–49	39	29.3	17	24.3
	50–59	17	12.8	6	8.6
	60+	1	0.8	1	1.4
Education level	Bachelor	97	72.9	54	77.1
	Master	14	10.5	6	8.6
	Ph.D.	13	9.8	6	8.6
	Others	9	6.8	4	5.7
Major	Medicine	10	7.5	6	8.6
	Nursing	72	54.1	41	58.6
	Nutrition	30	22.6	14	20.0
	Athletics	5	3.8	4	5.7
	Public health	3	2.3	0	0.0
	Others	13	9.8	5	7.1
Work place	University	15	11.3	11	15.7
	HyDiREC	78	58.6	45	64.3
	Health Center	40	30.1	14	20.0
Total		133	100.0	70	100.0

Ph.D.: Doctor of Philosophy; HyDiREC: Hypertension and Diabetes Registration and Education Center. * If the consistency ratio (CR) exceeded 0.2, data of respondents were excluded from the final analysis.

**Table 2 ijerph-18-00455-t002:** Relative importance by evaluation criteria of community-based intervention program to approach strategy for patients (individual level).

First Sub-Criteria	Third Sub-Criteria	Mean Score	Rank
Health behavior	Risk awareness for hypertension & diabetes (likelihood/severity)	6.23	2
	Cue to action for health behavior	5.99	4
	Self-efficacy	5.80	6
	Experience of socialization	5.39	14
	Healthy life-style (anti-smoking, reducing alcohol, nutrition, physical activities)	6.26	1
	Health examination participation rate (blood pressure/glucose test)	6.14	3
Behavior of health care utilization	Characteristic of medication use (dosage, frequency, drug form)	5.68	10
	Family supporting system/social network	5.62	11
	Patient-doctor relationship (family doctor etc.)	5.55	13
	Using possibility of information related hypertension & diabetes	5.62	11
	Assuring accessibility (economic/geographic/psychologic)	5.73	8
	Adequacy of community resource (workforce and facilities, delivery system)	5.70	9
	Well-timed health policy	5.77	7
	Patient satisfaction	5.99	4

**Table 3 ijerph-18-00455-t003:** Relative importance by evaluation criteria of community-based intervention program to approach strategy for healthcare system (community level).

First Sub-Criteria	Second Sub-Criteria	Third Sub-Criteria	Mean Score	Rank
Program infrastructure	Investment resources for building infrastructure	Feasibility of plan for demonstration program	5.89	16
		Adequacy of governmental budget for program	5.96	11
		Construct computer system for program	6.00	9
		Adequacy of region selected for demonstration program	5.53	28
		Adequacy of the committed institution selected	5.47	30
		Suitability of the cooperative institutes limited	5.37	36
	Participation system within community	Operation policy of demonstration program	5.53	28
		Build up the public-private partnership	5.95	12
		Voluntary participation level of community residents	6.06	6
		Linkage with community resources	6.01	8
Program process	Adequacy of center operation & service provided	Adequate selection & disposition of manpower	6.18	4
		Working continuity of manpower	6.23	2
		Specialty (empowerment) of manpower	6.27	1
		Time required for services (counselling/repayment cost)	5.66	21
		Adequacy of health education content (disease, nutrition, exercise)	6.19	3
		Adequacy of recall & remind service provided	5.64	23
		Adequacy of PR & campaign of demonstration program	5.75	19
		Adequacy of incentive provision for health behavior improvement	5.47	30
		Development & operation of region specialized program	5.73	20
		Evaluation & revision of computer system	5.92	15
	Adequacy of administrative procedures	Adequacy of registry process for patients	5.62	24
		Adequacy for finding the unmet needs people	5.40	35
		Adequacy of problem-solving method	5.56	26
		Legality for using the registry DB & personal information	5.55	27
		Adequacy of repayment process of costs	5.43	33
		Observance of the demonstration program guidance	5.42	34
Program outcome	Program output	Registry rate of patients	5.60	25
		Regular visiting rate to (healthcare) medical facilities of patients	5.83	17
		Rate of patients without treatment	5.80	18
		Pass rate of health education (disease, nutrition, exercise)	5.66	21
	Program performance	Service (quantity/quality) satisfaction	5.95	12
		Participation rate of private clinic	6.17	5
		Participation rate of pharmacy	6.03	7
		Adequacy of economic coverage (medical fee + drug cost)	5.93	14
		Adequacy of policy unsupported to 30~64 years	5.44	32
		Reducing the death rate of cardiovascular diseases (long-term effect)	5.99	10

**Table 4 ijerph-18-00455-t004:** Priority by each evaluation criteria for community-based intervention program with pairwise comparison.

Main Criteria	First Sub-Criteria	Second Sub-Criteria	Third Sub-Criteria	Priorities (Global Weights)	Importance Rank
Approach strategy for patients (individual level) (Global weight: 0.525, Importance rank: 1)	Health behavior (Global weight: 0.391, Importance rank: 2)	-	Risk awareness for hypertension & diabetes (likelihood/severity)	0.0318	5
			Cue to action for health behavior	0.0229	10
			Self-efficacy	0.0232	8
			Experience of socialization	0.0149	14
			Healthy life-style (anti-smoking, reducing alcohol, nutrition, physical activities)	0.0393	2
			Health examination participation rate (blood pressure/glucose test)	0.0343	4
	Behavior of health care utilization (Global weight: 0.461, Importance rank: 1)	-	Characteristic of medication use (dosage, frequency, drug form)	0.0152	13
			Family supporting system/social network	0.0226	11
			Patient-doctor relationship (family doctor etc.)	0.0232	8
			Using possibility of information related hypertension & diabetes	0.0202	12
			Assuring accessibility (economic/geographic/psychologic)	0.0246	6
			Adequacy of community resource (workforce and facilities, delivery system)	0.0243	7
			Well-timed health policy	0.0353	3
			Patient satisfaction	0.0471	1
Approach strategy for healthcare system (community level) (Global weight: 0.332, Importance rank: 2)	Program infrastructure (Global weight: 0.295, Importance rank: 1)	Investment resources for building infrastructure (Global weight: 0.420, Importance rank: 2)	Feasibility of plan for demonstration program	0.0050	31
			Adequacy of governmental budget for program	0.0064	23
			Construct computer system for program	0.0059	24
			Adequacy of region selected for demonstration program	0.0052	29
			Adequacy of the committed institution selected	0.0050	31
			Suitability of the cooperative institutes limited	0.0042	35
		Participation system within community (Global weight: 0.436, Importance rank: 1)	Operation policy of demonstration program	0.0045	34
			Build up the public-private partnership	0.0080	19
			Voluntary participation level of community residents	0.0115	15
			Linkage with community resources	0.0091	16
	Program process (Global weight: 0.272, Importance rank: 2)	Adequacy of center operation & service provided (Global weight: 0.672, Importance rank: 1)	Adequate selection & disposition of manpower	0.0001	50
			Working continuity of manpower	0.0058	25
			Specialty (empowerment) of manpower	0.0073	21
			Time required for services (counselling/repayment cost)	0.0040	37
			Adequacy of health education content (disease, nutrition, exercise)	0.0036	41
			Adequacy of recall & remind service provided	0.0039	40
			Adequacy of PR & campaign of demonstration program	0.0040	37
			Adequacy of incentive provision for health behavior improvement	0.0041	36
			Development & operation of region specialized program	0.0047	33
			Evaluation & revision of computer system	0.0056	28
		Adequacy of administrative procedures (Global weight: 0.249, Importance rank: 2)	Adequacy of registry process for patients	0.0021	49
			Adequacy for finding the unmet needs people	0.0023	48
			Adequacy of problem-solving method	0.0028	45
			Legality for using the registry DB & personal information	0.0028	45
			Adequacy of repayment process of costs	0.0034	42
			Observance of the demonstration program guidance	0.0034	42
	Program outcome (Global weight: 0.248, Importance rank: 3)	Program output (Global weight: 0.453, Importance rank: 1)	Registry rate of patients	0.0057	26
			Regular visiting rate to (healthcare) medical facilities of patients	0.0083	18
			Rate of patients without treatment	0.0080	19
			Pass rate of health education (disease, nutrition, exercise)	0.0086	17
		Program performance (Global weight: 0.430, Importance rank: 2)	Service (quantity/quality) satisfaction	0.0025	47
			Participation rate of private clinic	0.0057	26
			Participation rate of pharmacy	0.0040	37
			Adequacy of economic coverage (medical fee + drug cost)	0.0051	30
			Adequacy of policy unsupported to 30~64 years	0.0034	42
			Reducing the death rate of cardiovascular diseases (long-term effect)	0.0067	22

**Table 5 ijerph-18-00455-t005:** Priorities by alternative policy (alternative preference).

Alternative Policy	Priorities (Alternative Preference)
Improvement for treatment compliance	0.550
Enhancement for continuity of care	0.396

**Table 6 ijerph-18-00455-t006:** Gap analysis of alternative priorities (alternative preference) based on evaluation criteria.

Level	Evaluation Criteria	Improvement for Treatment Compliance	Enhancement for Continuity of Care	Gap
Main criteria	Approach strategy for patients	0.3719	0.2246	0.1473
	Approach strategy for healthcare system	0.1990	0.2011	−0.0021
First sub-criteria	Health behavior	0.1779	0.0950	0.0829
	Behavior of health care utilization	0.1940	0.1296	0.0644
	Program infrastructure	0.0633	0.0726	−0.0093
	Program process	0.0691	0.0785	−0.0094
	Program outcome	0.0734	0.0539	0.0195
Second sub-criteria	Investment resources for building infrastructure	0.0434	0.0313	0.0121
	Participation system within community	0.0320	0.0302	0.0018
	Adequacy of center operation & service provided	0.0276	0.0450	−0.0174
	Adequacy of administrative procedures	0.0173	0.0460	−0.0287
	Program output	0.0323	0.0216	0.0107
	Program performance	0.0382	0.0352	0.0030

## Data Availability

Data sharing is not applicable to this article.

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
