# Peer review of "Prioritizing Community-Based Intervention Programs for Improving Treatment Compliance of Patients with Chronic Diseases: Applying an Analytic Hierarchy Process"

_ijerph, 2021, doi:10.3390/ijerph18020455_

Round 1

Reviewer 1 Report

The Analytic Hierarchy Process was new to me; it seems it shares similarities with discrete choice experiment.  Still, even after reading carefully the methods and the results sections, I did not fully understand how the data collection and analysis were conducted, that is how the ranking and the weights were elicited from participants: was there direct comparison and choice between 2 alternatives each time, or was it independent rating.

The results are interesting and useful for health policymakers. However, the authors did not draw conclusions regarding the specialty needed for behavior/lifestyle changes.

Relatedly, it was interesting to read (in the discussion) that the authors think that in order to improve compliance by behavior change, there is need for support from the primary care physicians. However, physicians are not trained in behavior change. Are there other options? Other healthcare practitioners?

Under discussion – there is no need to re-iterate the epidemiological data of the first paragraph. It should be part of the introduction.

The sentence “‘the voluntary participation of community resident’ among the
 community participation system within program infrastructure as system (public health system)
 domain was found to be important” – unclear syntactically. Likewise ll. 386-389.

LL 397-400 seem to be based on professional ideology and less on data from the current analysis. Overall, the first limitation (ll 394-400) is uncelar.

The manuscript should end with a conclusion rather than with a limitation.

Author Response

Prioritizing Community-based Intervention Programs for Improving Treatment Compliance of Patients with Chronic Diseases: Applying an Analytic Hierarchy Process

Response to Reviewer 1:

We would like to thank the reviewer for your thoughtful comments. The comments have been helpful in revising our manuscript for improving the quality. We have incorporated changes related to all comments by 3 reviewers in the revised manuscript.

To facilitate a re-review of these issues, we first reproduced the reviewers’ comments, and then in the indented paragraphs we provide a summary of our response. Newly added sentences in the revised manuscript are shown in yellow or blue colors.

Comment 1: The Analytic Hierarchy Process was new to me; it seems it shares similarities with discrete choice experiment.  Still, even after reading carefully the methods and the results sections, I did not fully understand how the data collection and analysis were conducted, that is how the ranking and the weights were elicited from participants: was there direct comparison and choice between 2 alternatives each time, or was it independent rating.

Response to Comment 1: In accordance with your suggestion, we have adjusted the section of ‘materials and methods’ in the revised manuscript (p.4-7). Especially, we described more detailed information about data collection and priority analysis in LL 245-256.

Comment 2: The results are interesting and useful for health policymakers. However, the authors did not draw conclusions regarding the specialty needed for behavior/lifestyle changes.

Relatedly, it was interesting to read (in the discussion) that the authors think that in order to improve compliance by behavior change, there is need for support from the primary care physicians. However, physicians are not trained in behavior change. Are there other options? Other healthcare practitioners?

Response to Comment 2: In accordance with your suggestion, we have adjusted the section of ‘discussion’ in the revised manuscript (p.15-16). Especially, we described more detailed information about behavior/lifestyle changes in LL 484-534.

Comment 3: Under discussion – there is no need to re-iterate the epidemiological data of the first paragraph. It should be part of the introduction.

Response to Comment 3: In accordance with your suggestion (epidemiological data of diabetes in Korea), we have deleted this section of ‘discussion’ in the revised manuscript because of duplication in the introduction part, too (p.15).

Comment 4: The sentence “‘the voluntary participation of community resident’ among the
 community participation system within program infrastructure as system (public health system)
 domain was found to be important” – unclear syntactically. Likewise ll. 386-389.

Response to Comment 4: In accordance with your suggestion, we have adjusted the section of ‘discussion’ in the revised manuscript (p.17) in LL 568-570.

“health care system [89-90]. As seen in result of this research, ‘the voluntary participation level of community resident’ within program infrastructure as (public health system) domain was found to be important. Moreover, considering the ‘participation of private clinic and pharmacy’ which is basic..”

Comment 5: LL 397-400 seem to be based on professional ideology and less on data from the current analysis. Overall, the first limitation (ll 394-400) is uncelar.

Response to Comment 5: In accordance with your suggestion, we have repositioned this section of ‘discussion’ in the revised manuscript (p.16-17) in LL 515-521 and LL 574-577. Of course, we deleted this first section of limitation in LL 580-592.

“Although this study extracted and analyzed factors of the evaluation criteria on the relative importance for sustainable management of hypertension and diabetes, it had the following limitations. First, the response rate was high (75.7%), although contents of the questionnaire survey with the AHP method might be somewhat unfamiliar to health practitioners at 19 sites, unlike other general surveys. However, the representation of health project participants nationwide in 254 public health centers and the intrinsic value of selective bias could not be ignored. Thus, there may be limitations in interpretation and utilization of our results. Second, as a result of simplifying the hierarchy of evaluation criteria by focusing on securing consistency in the evaluation through the prior systematic review and Delphi technique, there was a limit to deriving concrete and in-depth analysis.”

Comment 6: The manuscript should end with a conclusion rather than with a limitation.

Response to Comment 6: In accordance with your suggestion, we have inserted the section of ‘conclusion’ in the revised manuscript (p.18) in LL 594-607.

  1. Conclusion

We analyzed the relative importance of evaluation criteria using our proposed AHP model for intervention and management of chronic diseases in Korea. Using the evaluation of the relative importance of the AHP model for sustainable management of hypertension and diabetes, all participants in our survey seemed to view patients’ preventive strategy more importantly given clinical features of hypertension and diabetes. Also, the domain of appropriateness of center operation and service provision under the program process evaluation was highly ranked as a healthcare system strategy. In conclusion, for sustainable management of hypertension and diabetes within regional communities, not only service provision through selection and continuous work of capable staff, but also the appropriateness of health education about health related behavior that can improve a patient’s satisfaction and treatment adherence are important. Future research should focus on expanding survey subjects to primary care physicians and even policymakers of central government for the proper use of this AHP model.

Reviewer 2 Report

The introduction section does not provide sufficient background to understand the aim of the study.

It is not clear how two alternative approaches were selected for the analysis with the AHP model.  

The Methods section lack clear information about characteristics of subjects participating in the expert groups. The characteristics of the respondents are also lacking? It is not clear what “health practitioners working in registration and education centres for hypertension and diabetes…” means.

Apart from this,  the justification why the survey was performed in such group and no respondents from outside mentioned centres were included, e.g. general practitioners from the region.

Although the results of the analysis may be compelling, the way the results are presented is highly discouraging and not transparent. I would recommend making an effort to provide the results in a more concise form. Apart from this, it is rather difficult to understand reported results if the tables and figures extend on several pages. I would suggest starting combining the tables and using a decidedly smaller font to decrease the size of tables. Now, an attempt of understanding what is included in specific tables, and where scores/values provided come from, is a painful experience. Authors should also take care of providing informative titles to the tables.

Figure 5 is amazing; a reader can spend long hours to understand from what higher-level categories, level III criteria presented in this figure come. It is not clear what is the meaning of the list of criteria without any values on page 10  within this figure? Part of bars is not assigned with criteria.

Apart from this, the figure seems to be useless, as the same information is provided in table 3. Preceding the table 3 with figure 5 increase the feeling of chaos when going through the results section.

What is the sense for adding “huge” chart with two 3D bars in figure 4 and provide the same limited information in table 4?

Finally, it seems that table 1, 2 and 3 can be integrated after an appropriate description of column headings describing mean scores, their importance ranks, priorities and relevant importance ranks. It is not clear why authors multiplied the number of tables with the failure of clear presentation of relevant indicators

Discussion is only a recapitulation of the Results section, but practically no references to prior studies are provided. It would be interesting to see how the results obtained by authors agree or disagree with the priorities in chronic care developed/assumed by other teams.

As the scope of results is rather extensive, the conclusions drawing the reader’s attention to the main finding at the end of the paper would be useful.  

The text should be checked up and polished by a native speaker. From time to time, one can find quite peculiar expressions. For example, in line 345-346: “to be more highly considered important” or the sentence in lines 355-356: “program. each ranked highest in priority”. Some sentences are simply indigestible, e.g. lines 357-369.

Author Response

Prioritizing Community-based Intervention Programs for Improving Treatment Compliance of Patients with Chronic Diseases: Applying an Analytic Hierarchy Process

Response to Reviewer 2:

We would like to thank the reviewer for your thoughtful comments. The comments have been helpful in revising our manuscript for improving the quality. We have incorporated changes related to all comments by 3 reviewers in the revised manuscript.

To facilitate a re-review of these issues, we first reproduced the reviewers’ comments, and then in the indented paragraphs we provide a summary of our response. Newly added sentences in the revised manuscript are shown in yellow or blue colors.

Comment 1: The introduction section does not provide sufficient background to understand the aim of the study.

Response to Comment 1: In accordance with your suggestion, we have made up for the weak points about background to the aim of the study in the revised manuscript (p.2-4). Especially, we divided the introduction into two sessions and described it in more detail in LL 44-142.

Comment 2: It is not clear how two alternative approaches were selected for the analysis with the AHP model.  

Response to Comment 2: In accordance with your suggestion, we have inserted this section of ‘materials and methods’ in the revised manuscript (p.4-5) in LL 162-182.

Comment 3: The Methods section lack clear information about characteristics of subjects participating in the expert groups. The characteristics of the respondents are also lacking? It is not clear what “health practitioners working in registration and education centres for hypertension and diabetes…” means. Apart from this,  the justification why the survey was performed in such group and no respondents from outside mentioned centres were included, e.g. general practitioners from the region.

Response to Comment 3: In accordance with your suggestion, we have inserted this section of ‘materials and methods’ and ‘result’ in the revised manuscript (p.5-8) in LL 184-296.

Comment 4: Although the results of the analysis may be compelling, the way the results are presented is highly discouraging and not transparent. I would recommend making an effort to provide the results in a more concise form. Apart from this, it is rather difficult to understand reported results if the tables and figures extend on several pages. I would suggest starting combining the tables and using a decidedly smaller font to decrease the size of tables. Now, an attempt of understanding what is included in specific tables, and where scores/values provided come from, is a painful experience. Authors should also take care of providing informative titles to the tables.

Response to Comment 4: In accordance with your suggestion, we have adjusted this section of ‘result’ in the revised manuscript (p.9-15) in LL 308-462. We made as more concise forms that all tables could be included within one page, and the title was also modified to suit the content.

Comment 5: Figure 5 is amazing; a reader can spend long hours to understand from what higher-level categories, level III criteria presented in this figure come. It is not clear what is the meaning of the list of criteria without any values on page 10 within this figure? Part of bars is not assigned with criteria. Apart from this, the figure seems to be useless, as the same information is provided in table 3. Preceding the table 3 with figure 5 increase the feeling of chaos when going through the results section. What is the sense for adding “huge” chart with two 3D bars in figure 4 and provide the same limited information in table 4?

Response to Comment 5: In accordance with your suggestion, we have adjusted this section of ‘result’ in the revised manuscript (p.12-13) in LL 395-406. First, we deleted figure 3 and figure 4. We made as more concise forms that pre-figure 5 (changed as figure 3) could be included within one page, and the title was also modified to suit the content.

Comment 6: Finally, it seems that table 1, 2 and 3 can be integrated after an appropriate description of column headings describing mean scores, their importance ranks, priorities and relevant importance ranks. It is not clear why authors multiplied the number of tables with the failure of clear presentation of relevant indicators.

Response to Comment 6: In accordance with your suggestion, we have adjusted this section of ‘result’ in the revised manuscript (p.9-12) in LL 337-397. We unified relevant indicators of each column headings in table 2, 3 and 4. However, we didn’t integrate these tables because the measurement method and its meaning are different.

Comment 7: Discussion is only a recapitulation of the Results section, but practically no references to prior studies are provided. It would be interesting to see how the results obtained by authors agree or disagree with the priorities in chronic care developed/assumed by other teams.

Response to Comment 7: In accordance with your suggestion, we have adjusted the section of ‘discussion’ in the revised manuscript (p.15-16). Especially, we described more detailed information about alternative policy of this study including the priorities in chronic care in LL 466-542.

Comment 8: As the scope of results is rather extensive, the conclusions drawing the reader’s attention to the main finding at the end of the paper would be useful.  

Response to Comment 8: In accordance with your suggestion, we have left this section of ‘discussion’ untouched in the revised manuscript in LL 543-572 (p.17). However, we changed ‘suggestions’ to ‘policy implications’ in LL 543.

Comment 9: The text should be checked up and polished by a native speaker. From time to time, one can find quite peculiar expressions. For example, in line 345-346: “to be more highly considered important” or the sentence in lines 355-356: “program. each ranked highest in priority”. Some sentences are simply indigestible, e.g. lines 357-369.

Response to Comment 9: In accordance with your suggestion, we have taken the proofreading by a native speaker. All loose sentences including LL 533-535 (“program. each ranked highest in priority”) and LL 596-599 (“to be more highly considered important”), etc. deleted or adjusted.

Reviewer 3 Report

Review of

Prioritizing on the Community-based Intervention Program for Improving Patient Compliance with Chronic Diseases: Applying the Analytic Hierarchy Process

  1. Title and Abstract: They reflect the main purpose of the manuscript precisely. The authors tried to contribute to this field of research
  2. Abstract: In the abstract the authors mention that the brainstorming and Delphi techniques were used. Because these techniques are no longer mentioned in the paper, please explain how you used these techniques
  3. Results: This Section is called Results not Empirical results.

Figure 3. Analyzing the relative importance of evaluation criteria using the APH method (Level 1) - contains information that can be represented in a table or in text, the graphical representation is not relevant. Figure 5. - Analyzing the relative importance of evaluation criteria using the AHP method (Level III) - the figure must be remade in another form, because in this form the presented results cannot be interpreted

  1. Discussions: This section is the 4th . The discussions may be combined with results. authors need to discuss all the presented results and the correlation between them and with data from other literature, without repeating the sentences from the Results section.

The authors need to complete the Discussions section with the comparison of the obtained data with the similar data obtained in other similar studies.  Future research directions may also be mentioned.

  1. Conclusions: This section is mandatory and should provide readers with a brief summary of the main conclusions. The authors need to formulate the conclusions, after discussions, according to the Instructions for authors.
  2. References: The authors need to check the list of references and correct it according to the Instructions for authors.
  3. In the Introduction there are several sentences (between 67-78 lines) that are overlapping with this bibliographic source biomedcentral.com. The authors need to rephrase the sentences and include this bibliographic source in the list of references.

In the article there are several sentences (between 102-107, 115-117, 150-154, 157-170, 176-181 lines) that are overlapping with this bibliographic source Taeksoo Shin, Chun-Bae Kim, Yang-Heui Ahn, Hyo-Youl Kim et al. "The comparative evaluation of expanded national immunization policies in Korea using an analytic hierarchy process",Vaccine, 2009. The authors need to rephrase the sentences.

Author Response

Prioritizing Community-based Intervention Programs for Improving Treatment Compliance of Patients with Chronic Diseases: Applying an Analytic Hierarchy Process

Response to Reviewer 3:

We would like to thank the reviewer for your thoughtful comments. The comments have been helpful in revising our manuscript for improving the quality. We have incorporated changes related to all comments by 3 reviewers in the revised manuscript.

To facilitate a re-review of these issues, we first reproduced the reviewers’ comments, and then in the indented paragraphs we provide a summary of our response. Newly added sentences in the revised manuscript are shown in yellow or blue colors.

Comment 1:

  1. Title and Abstract: They reflect the main purpose of the manuscript precisely. The authors tried to contribute to this field of research

Response to Comment 1: In accordance with other reviewer suggestion, we have taken the proofreading by a native speaker in the all section of this manuscript including title and abstract in LL 1-31 (p.1).

Comment 2:

  1. Abstract: In the abstract the authors mention that the brainstorming and Delphi techniques were used. Because these techniques are no longer mentioned in the paper, please explain how you used these techniques.

Response to Comment 2: In accordance with your suggestion, we have adjusted the section of ‘materials and methods’ in the revised manuscript (p.5-6). Especially, we described more detailed information about the section of decision criteria and priority analysis in LL 183-237.

Comment 3:

  1. Results: This Section is called Results not Empirical results.

Figure 3. Analyzing the relative importance of evaluation criteria using the APH method (Level 1) - contains information that can be represented in a table or in text, the graphical representation is not relevant. Figure 5. - Analyzing the relative importance of evaluation criteria using the AHP method (Level III) - the figure must be remade in another form, because in this form the presented results cannot be interpreted

Response to Comment 3: In accordance with your suggestion, we changed ‘Empirical results’ to ‘Results’ in LL 271 (p.7).

Also, in accordance with your and other reviewer suggestion, we have adjusted this section of ‘result’ in the revised manuscript (p.9-13) in LL 308-406. First, we deleted figure 3 and figure 4. We made as more concise forms that pre-figure 5 (changed as figure 3) could be included within one page, and the title was also modified to suit the content.

Comment 4:

  1. Discussions: This section is the 4th . The discussions may be combined with results. authors need to discuss all the presented results and the correlation between them and with data from other literature, without repeating the sentences from the Results section.

The authors need to complete the Discussions section with the comparison of the obtained data with the similar data obtained in other similar studies.  Future research directions may also be mentioned.

Response to Comment 4: In accordance with your and other reviewer suggestion, we have adjusted the section of ‘discussion’ in the revised manuscript (p.15-18). Especially, we described more detailed information without repeating the sentences from the Results section in LL 466-542.

Comment 5:

  1. Conclusions: This section is mandatory and should provide readers with a brief summary of the main conclusions. The authors need to formulate the conclusions, after discussions, according to the Instructions for authors.

Response to Comment 5: In accordance with your and other reviewer suggestion, we have inserted the section of ‘conclusion’ in the revised manuscript (p.18) in LL 594-607.

  1. Conclusion

We analyzed the relative importance of evaluation criteria using our proposed AHP model for intervention and management of chronic diseases in Korea. Using the evaluation of the relative importance of the AHP model for sustainable management of hypertension and diabetes, all participants in our survey seemed to view patients’ preventive strategy more importantly given clinical features of hypertension and diabetes. Also, the domain of appropriateness of center operation and service provision under the program process evaluation was highly ranked as a healthcare system strategy. In conclusion, for sustainable management of hypertension and diabetes within regional communities, not only service provision through selection and continuous work of capable staff, but also the appropriateness of health education about health related behavior that can improve a patient’s satisfaction and treatment adherence are important. Future research should focus on expanding survey subjects to primary care physicians and even policymakers of central government for the proper use of this AHP model.

Comment 6:

  1. References: The authors need to check the list of references and correct it according to the Instructions for authors.

Response to Comment 6: In accordance with your suggestion, we have re-checked this section of ‘the list of references’ in the revised manuscript (p.18-26) in LL 614-863.

Comment 7:

  1. In the Introduction there are several sentences (between 67-78 lines) that are overlapping with this bibliographic source biomedcentral.com. The authors need to rephrase the sentences and include this bibliographic source in the list of references.

In the article there are several sentences (between 102-107, 115-117, 150-154, 157-170, 176-181 lines) that are overlapping with this bibliographic source. Taeksoo Shin, Chun-Bae Kim, Yang-Heui Ahn, Hyo-Youl Kim et al. "The comparative evaluation of expanded national immunization policies in Korea using an analytic hierarchy process",Vaccine, 2009. The authors need to rephrase the sentences.

Response to Comment 7: In accordance with your suggestion, we have rephrased these overlapping sentences (between 75-86, 132-134, 140-142, 212-215, 219-222, 232-233, 240-245 lines) with the bibliographic sources in each list of references (including Shin et al. study in Vaccine) in the revised manuscript (p.2-6).

Round 2

Reviewer 1 Report

It seems the manuscript did NOT undergo language editing. There are still many places where the language is inadequate (e.g. “the survey was progressed from…”).

Ll. 64-66 on p. 2 are syntactically incomprehensible. Some errors occurred at the correction stage.

L 266 (p. 7) – what is “senior health keeper”?

l. 307 – “9 to 1 to 9” – unclear.

Please add in limitations the absence of patient and public involvement.

“Health education” is only a first stage in lifestyle changes. The conclusion should emphasize the outcome of behavior change by persons with diabetes and hypertension and not producing health education. The survey results are still not reflected in the discussion and conclusions; they are well presented in the abstract.

Author Response

Prioritizing Community-based Intervention Programs for Improving Treatment Compliance of Patients with Chronic Diseases: Applying an Analytic Hierarchy Process

Response to Reviewer 1:

We would like to thank a lot for your meticulous review. Your comments are reflected in our manuscript for improving the quality. We have incorporated changes related to all comments by reviewers in the second revised manuscript.

To facilitate a re-review of these issues, we first reproduced the reviewers’ comments, and then in the indented paragraphs we provide a summary of our response. Adjusted sentences in the second revised manuscript are shown in yellow. Newly added sentences in the second revised manuscript are shown in blue colors.

Comment 1: Extensive editing of English language and style required. It seems the manuscript did NOT undergo language editing. There are still many places where the language is inadequate. (e.g. “the survey was progressed from…”). Ll. 64-66 on p. 2 are syntactically incomprehensible. Some errors occurred at the correction stage. l. 307 – “9 to 1 to 9” – unclear.

Response to Comment 1: In accordance with your suggestion, we took twice the proofreading by same native speaker.

  • 62-68 on p. 2

Empirically, Marques-Vidal et al. [13] have performed a systematic review and indicated that, although the ‘rule of halves’ are no longer applied for screening and treating hypertensive patients in industrialized countries, it might still be valid for early detecting and caring for hypertension in developing countries and for effective antihypertensive drug treatment in all countries.

In South Korea, due to its rapid economic growth and aging since 1990s, chronic diseases such as hypertension and diabetes instead of infectious diseases have surged to bring a huge national medical cost. Amid epidemiological transmission, this rule of halves has also been shown to be valid in Korea's community-based intervention program for controlling hypertension [14].

  • 244 – “9 to 1 to 9” – unclear.

from 9 to 1 (9 points in both directions)

  • 254-255 on p. 7

the survey was conducted from April 10, 2014 to May 3, 2014.

Comment 2: L 266 (p. 7) – what is “senior health keeper”?

Response to Comment 2: In accordance with your suggestion, we have adjusted the words of ‘materials and methods’ and ‘discussion’ in the revised manuscript (p.5, p.17).

senior community health volunteers

Comment 3: Please add in limitations the absence of patient and public involvement.

Response to Comment 3: In accordance with your suggestion, we have added two sentences of ‘discussion’ in the revised manuscript (LI 583-586, p.17-18).

Second, our AHP questionnaire survey was not available to hypertensive and diabetes patients and the general public. This is because AHP questionnaire contents were very high difficulty index and large in its quantity, so they were not easily written.

Comment 4: “Health education” is only a first stage in lifestyle changes. The conclusion should emphasize the outcome of behavior change by persons with diabetes and hypertension and not producing health education. The survey results are still not reflected in the discussion and conclusions; they are well presented in the abstract.

Response to Comment 4: In accordance with your suggestion, we have added two sentences of ‘conclusion’ in the revised manuscript (LI 605-607, p.18). Fourteen references were already added to the first revision process for reflecting the results in discussions and conclusions.

Health education is only a first stage in lifestyle modification. Continuous health education program for patients with diabetes and hypertension within a community needed to their behavior change accordingly.

Reviewer 2 Report

The authors introduced most of the suggested changes. Now the manuscript is more transparent and understandable to the reader.

Author Response

Prioritizing Community-based Intervention Programs for Improving Treatment Compliance of Patients with Chronic Diseases: Applying an Analytic Hierarchy Process

Response to Reviewer 2:

Comment: The authors introduced most of the suggested changes. Now the manuscript is more transparent and understandable to the reader. English language and style are fine/minor spell check required.

Response to Comment: In accordance with your suggestion, we took twice the proofreading by same native speaker.

We would like to thank a lot for your meticulous review. Your comments are reflected in our manuscript for improving the quality. We have incorporated changes related to all comments by reviewers in the second revised manuscript.

To facilitate a re-review of these issues, we first reproduced the reviewers’ comments, and then in the indented paragraphs we provide a summary of our response. Adjusted sentences in the second revised manuscript are shown in yellow. Newly added sentences in the second revised manuscript are shown in blue colors.

Reviewer 3 Report

I accept the authors' corrections and in my opinion the article can be published in this form

Author Response

Prioritizing Community-based Intervention Programs for Improving Treatment Compliance of Patients with Chronic Diseases: Applying an Analytic Hierarchy Process

Response to Reviewer 3:

Comment: I accept the authors' corrections and in my opinion the article can be published in this form.

We would like to thank a lot for your meticulous review. Your comments are reflected in our manuscript for improving the quality. We have incorporated changes related to all comments by reviewers in the second revised manuscript.

To facilitate a re-review of these issues, we first reproduced the reviewers’ comments, and then in the indented paragraphs we provide a summary of our response. Adjusted sentences in the second revised manuscript are shown in yellow. Newly added sentences in the second revised manuscript are shown in blue colors.
